# Environmental Consciousness, Purchase Intention, and Actual Purchase Behavior of Eco-Friendly Products: The Moderating Impact of Situational Context

**DOI:** 10.3390/ijerph20075312

**Published:** 2023-03-29

**Authors:** Nayeon Kim, Kyungtag Lee

**Affiliations:** School of Management, Yeungnam University, Gyeongsan 38541, Republic of Korea

**Keywords:** green product, environmental consciousness, purchase intention, actual purchase behavior, situational context

## Abstract

Recently, environmental issues have become major social concerns, and consumers are becoming increasingly aware of environmental matters; however, they remain hesitant to purchase eco-friendly products. This study examined consumers’ environmental consciousness as a factor influencing the purchase of eco-friendly products, and investigated situational factors that induce hesitancy in purchasing eco-friendly products. We studied the moderating effects of these factors with regard to ease of purchase and eco label credibility. Our research model is validated using data from 220 consumers with experience in purchasing eco-friendly products in Korea. For the data analysis, we used SPSS 22.0 and AMOS 22.0 to perform confirmatory factor analysis and SEM. The specific verification results are as follows. First, environmental interest did not significantly impact the purchase intention of eco-friendly products. Second, consumers’ environmental knowledge and consumer effectiveness perception both had a significant impact on the purchase intention of eco-friendly products. Third, the intention to purchase eco-friendly products significantly impacted the purchase behavior of eco-friendly products. In addition, the results of this study show that ease of purchase and eco label credibility have moderating effects on the relationship between purchase intention and purchase behavior. This study results contribute to the eco-friendly consumption literature by explaining the intention–behavior gap. This study also show that eco-friendly consumption can be stimulated through raising eco label credibility and ease of purchase. The findings have theoretical implications for understanding the factors that affect consumers’ intentions of and behavior toward eco product purchases, and practical implications for how to stimulate environmental consumer behavior.

## 1. Introduction

Recently, owing to the increase in particulates matters, frequently recurring instances of abnormal weather, and recycling issues, consumers have acquired an increasing awareness of environmental problems and interest in addressing them. In the past, consumers focused mainly on eco-friendly activities, such as recycling to solve environmental problems, but were relatively indifferent to the production and consumption of products that pollute the environment. However, nowadays, they have begun to oppose the production of products that pollute the environment and actively highlight the government’s passive attitude toward saving the environment. They practice environmentally friendly behaviors and prefer the consumption of eco-friendly products [1].

Consequently, many studies have investigated the factors that impact consumer behavior to purchase eco-friendly products—mainly, consumers’ environmental values, attitude, knowledge, product prices, and awareness [2,3,4,5]. Roberts [6] argues that environmental awareness is important to bridge the gap between environmental issues and sustainable behavior, and according to Joshi and Rahman [7], consumers’ high interest in environmental and social issues induces eco-friendly purchasing behavior, and is considered the main motivation. Consumers who are interested in environmental and ethical issues said that they prefer to purchase eco-friendly products [8]. However, research on the relationship between consumers’ environmental consciousness and eco-friendly product purchase behavior is scarce. Existing studies on environmental consciousness focus on eco-friendly behavior, but direct product purchase by consumers has not been explored.

In general, most of the consumers who are sensitive to environmental degradation support eco-friendly products; however, their support does not translate into actual action [9]. According to the 2020 Korea Procter & Gamble (P&G) survey of 4000 Korean consumers on purchasing eco-friendly products, 82.2% of all respondents reported that they were willing to purchase eco-friendly products; however, only 25.5% of the respondents actually purchased eco-friendly products [10]. The results suggest that willingness itself is unlikely to lead to the actual purchase of eco-friendly products. Moreover, the results show that, despite having environmentally friendly intentions, consumers are skeptical about purchasing eco-friendly products. In other words, people who are concerned about the environment do not necessarily buy and consume eco-friendly products. However, previous studies have measured the behavioral dimension related to purchasing eco-friendly products as purchase intention [11,12]. In fact, studies in consumer behaviors have usually viewed the intention as the same or at least highly correlated with the actual behavior [13]. The area of purchase intention and buying behavior gaps in consumers purchasing environmentally sustainable products has been extensively studied in the past literature [1,5,7,14,15,16]. However, there still exists a gap between the intention and the actual behavior.

In order to solve this problem, it is necessary to narrow the gap between green consumption intentions and actual behavior. It has also been proposed that more research on the discrepancy between purchase intention and purchase behavior should be performed [17]. However, there is a paucity of the literature examining the moderating effects of the gap between purchase intention and purchase behavior on green product consumption. Kaur and Bhardwaj evaluated the moderating influence of a proxy measure of actual control on the purchase intention–action gap and showed that it positively moderated the relationship between purchase intention and purchase behavior [16]. Joshi and Rahman [7] reviewed studies with regard to situational factors as barriers between consumers’ purchase intention and purchase behavior, and emphasized that future research on situational factors should continue. Grimmer et al. [18] revealed that the mediating effect of purchase intention and the moderating effect of situational factors appeared in the relationship between the purchase intention and purchase behavior of ethical products, and suggested that more research be conducted on the relationship between purchase intention and purchase behavior. 

Therefore, this study aims to examine the effect of each of these dimensions on eco-friendly product purchase behavior by dividing environmental consciousness into environmental knowledge, environmental interest, and consumer effectiveness perception. Moreover, we also investigate whether the situational context factors control the relationship between the purchase intention and purchase behavior of eco-friendly products. Through this study, the validity of the arguments made by existing studies can be confirmed, and practical implications related to eco-friendly products can be gained. 

In Section 2, we introduce our perspective and hypotheses. Next, we describe the research methodology. Then, we specify structural model and report empirical results. The last section discusses this study’s implications and limitations and then provides suggestions for future research. 

## 2. Literature Review and Hypothesis Development

### 2.1. Theory of Planned Behaviors

Since Ajzen and Fishbein’s theory of planned behavior (TPB) was first presented, it has been widely used to understand various human behaviors. They argued that behavior is observable and that behavior is determined by the intention to perform it [13]. Ajzen and Fishbein continued to pay attention to the factors influencing behavior and revealed the relationship between factors such as belief, attitude, and intention, as well as subjective norms and perceived behavioral control. Perceived behavioral control was said to influence intention, and at the same time regulate the relationship between intention and behavior. They called this relationship the reasoned action (TRA). In reviewing various areas of empirical evidence for the TRA or TPB model, Ajzen and Fishbein [13] acknowledged that there may be an intention–action gap, which they refer to as “literal inconsistency.”

The central logic of many studies on green consumption revolves around the theory of TPB [12]. Several studies on green consumption have used TRA or TPB, but some have focused on intention or behavior only [2,6,7,8]. Green attitudes and intentions have often been found to influence actual behavior toward green consumption, but an attitude–behavior gap still appears to exist. This is often the case when consumers show favorable attitudes or intend to behave in an environmentally friendly way, but do not actually act green [8,12].

### 2.2. Environmental Consciousness

Environmental consciousness was defined as a specific psychological factor related to an individual’s propensity to participate in eco-friendly behavior [19]. Environmental consciousness refers to “psychological factors that determine consumer propensity for eco-friendly behavior” [20]. It is the willingness to become aware of environmental problems, to support efforts to solve environmental problems, and to personally commit and act to solve these problems [21]. The concepts of environmental consciousness, which have been dealt with in preceding studies, mainly include an awareness of environmental problems, interest in, attitudes, and opinions on environmental problems, and are explained as an awareness to prevent and improve environmental pollution and damage that occur as a result of human activities. Environmental consciousness is defined in various ways and is dealt with in many studies as an intrinsic factor influencing an individual’s eco-friendly consumption behavior.

It was argued that environmental consciousness has a multidimensional structure composed of cognitive, attitudinal, and behavioral factors [2]. Environmental consciousness was divided into knowledge, attitude, recycling behavior type, recycling degree, and participation activities to prevent environmental destruction. Roberts [6] classified the environmental awareness dimension into consumer efficiency awareness and environmental interest. Sharmar and Kesherwani [22] divided the dimension of environmental consciousness into four categories: environmental value, attitude, knowledge, and motivation. Environmental value is the value of nature and nature conservation, and environmental knowledge is regarded as the knowledge of environmental issues. 

Environmental concern is the degree to which people are aware of and willing to support efforts to address environmental problems or personally contribute to solutions [23]. Schultz [24] classified environmental interest into three dimensions: egoistic concerns, altruistic concerns, and ecological-centered environmental concerns. Egoistic concerns refer to an interest in environmental issues related to one’s own health, future, or lifestyle. Altruistic concerns imply an interest in environmental issues related to everyone, including the community, children, and the future. Ecological-centered environmental concerns are the interest of environmental issues related to plants, animals, marine life, and birds.

Environmental knowledge can be defined as possessing facts about the natural environment and major ecosystems and a general knowledge of the relationship between people and the environment. It can also include what people know about the environment, key environmental or environmental relationships, recognition of the “whole system of the environment,” and knowing the responsibilities of stakeholders necessary for sustainable development [25]. Environmental knowledge is considered an approach to address environmental and social problems as consumers’ perceived knowledge of environmental and social problems [26]. Frick, Kaiser, and Wilson [27] classified environmental knowledge into system knowledge and behavior-related knowledge. The former implies the understanding of the natural state of the ecosystem and its processes; the latter refers to the knowledge of the actions that consumers can perform to have an impact on environmental issues.

Perceived consumer effectiveness (PCE) is a measure of the entity’s judgment on whether individual consumers themselves can influence environmental resource problems. It can be defined as the extent to which individual consumers believe they can contribute to address environmental problems through personal efforts and daily consumption behavior [6]. He and Zhan [28] defined it as the extent to which consumers believe that adopting eco-friendly cars can help reduce the negative impact of their vehicle usage on the environment. Ellen, Wiener, and Walgren [29] defined it as the degree to which one believes that one’s efforts or actions for the environment can make a difference in addressing environmental problems. Additionally, it has been shown that PCE on environmental issues is continuously linked to socially recognized attitudes, but is distinct from environmental issues or attitudes and makes a unique contribution to the prediction of environmentally conscious behaviors, such as eco-friendly purchasing. Additionally, if an individual believes that environmental problems can be addressed by a particular activity, this belief has a significant impact on the individual’s willingness to perform that activity.

### 2.3. Situational Context

Studies on consumer purchasing behavior have argued that many factors can influence buyers’ purchasing decisions. Carrington et al. [14] studied the reasons why consumers have ethical purchase intentions in their daily lives but which do not lead to purchase behavior. There were four main reasons. It was revealed that: (1) prioritization of ethical issues, (2) formation of plans/habits, (3) will and sacrifice, and (4) shopping behavior and situational factors all influence the gap between purchase intention and action. Gleim et al. [30] identified price, quality, professionalism, reliability, availability, apathy toward the environment, and brand loyalty as factors that hindered the purchase of eco-friendly products. Among them, price, a weak perception of product quality, trust, and ease of purchase were the largest obstacles. Hwang and Chung [31] reported that store quality, price perception, and corporate social responsibility beliefs were major antecedents of purchasing behavior. However, only very limited research has used mediating and moderating mechanisms, such as product quality and price sensitivity, in the link between consumer perceptions and behavior [32]. Shamsi et al. [33] and Molinillo et al. [34] suggested that more variables should be explored to provide additional insight into consumers’ perceptions and behavior toward organic food consumption. 

Carrington et al. [35] stated that the effect of purchase intention on actual behavior in purchasing ethical products is affected by factors that serve as barriers or catalysts. They used the context as a group variable to grasp the relationship between intention and action. It was argued that the existence of a positive situational context acts as a catalyst in the conversion of plans into actions. Grimmer et al. [18] investigated the scope of situational factors in the intention to purchase eco-friendly products leading to actual purchase behavior; however, it was noted that the factors of the situation are quite broad. They examined the context in eight different categories: price, distance, product availability, ease of purchase, time, effort, inconvenience, and purchasing possibility. Joshi and Rahman [7] identified situational factors that serve as barriers to the relationship between purchase intention and purchase behavior. In the context of the situation, there are price sensitivity, ease of purchase, norms, product attributes, product quality, store-related attributes, brand image, environmental labeling, certification, and other situational factors. Therefore, in this study, two variables: ease of purchase and eco label credibility, were selected as situational context variables based on previous studies.

#### 2.3.1. Ease of Purchase

The ease of purchase saves time and physical and mental energy required when shopping [36]. Vermeir and Verbeke [37] stated that ease of purchase is related to the availability of sustainable products related to consumer behavior control, which implies difficulty in obtaining or consuming specific products. Consumers stated that it was difficult to purchase products because, although they had high motivation to consume eco-friendly products, these products had a low ease of use. This problem is related to the lack of retail stores or product markets that sell these products, which leads to irregularities and a lack of convenience desired by consumers.

#### 2.3.2. Eco Label Credibility

The eco label provides identifiable marketing tools to communicate the environmentally friendly and socially desirable characteristics of a product to consumers [38]. Eco labels are known to improve consumer response to both green advertising and brands, and are considered to serve as objective guarantees for the environmental information of products. The trust in eco labels simplifies information retrieval and improves consumer decision-making. Gleim et al. [30] considered that trust is significant when considering the purchase of eco-friendly products, and a lack of trust in eco-friendly products cannot have a positive effect on eco-friendly consumers. Accordingly, it is judged that consumers’ trust in eco labels as an information source significantly impacts their decision to purchase eco-friendly products.

### 2.4. Research Model and Hypothesis

Based on the environmental consciousness mentioned by Joshi and Rahman [7], this study aimed to examine the effects of environmental consciousness on eco-friendly product purchase intention and the adjustment effect of label credibility. The research model of this study is shown in Figure 1.

#### 2.4.1. Environmental Consciousness and Intention to Purchase Eco Products

Many of the previous literature studies have reported that the effect of environmental awareness on purchasing products, such as organic food, is insignificant, and thus the influence of environmental awareness on the intention to purchase organic food has been underestimated in the existing literature [32]. Interest was also said to be an important factor that can affect consumers’ perceptions when purchasing eco-friendly products, such as organic food [39]. 

Environmentally conscious people often consider the environmental impact when purchasing products [35]. As such, they tend to buy organic food because they may perceive it to be safer, healthier, and less adversely impactful to the environment and eco systems [40]. Environmentally conscious consumers use more eco-friendly products than less environmentally friendly consumers [41]. Consumers’ high concern in environmental and social issues and the functional and eco-friendly characteristics of products are the main motivators for inducing eco-friendly purchasing behavior [7]. Additionally, Cottrell [42] argued that environmental concerns are a reasonable predictor for environmental behavioral intentions. The concern for the environment has a direct effect on purchased intention in an environmentally sustainable way [43]. Kim and Choi [1] stated that environmentalism does not impact collectivism, but influences eco-friendly purchasing behavior.

**Hypothesis** **1** **(H1).***Environmental concerns will have a positive effect on the purchase intention of eco-friendly products*.

As knowledge reflects the cognitive aspects of humans, environmental knowledge is considered a crucially meaningful factor that influences individuals to practice sustainable consumption [44]. Previous studies suggested that knowledge and perceptions of environmental issues can influence consumers’ purchase intentions for green products [20,45]. Mostafa [46] stated that practical environmental knowledge is necessary to take appropriate measures for ecological protection and that higher environmental knowledge is more likely to exhibit sustainable purchasing behavior. According to Maichum et al. [47], environmental knowledge has a positive effect on the purchase intention of green products.

**Hypothesis** **2** **(H2).***Environmental knowledge will have a positive effect on the purchase intention of eco-friendly products*.

Perceived consumer effectiveness (PCE) has been identified as a significant variable related to socially responsible behavior [7]. Perceived consumer effectiveness (PCE) on environmental issues is distinct from environmental concerns or attitudes and is an important factor in predicting environmentally conscious behaviors, such as green purchasing [29]. People with high levels of perceived consumer effectiveness (PCE) are more likely to engage in environmental action to alleviate their concerns about the environment [30]. In particular, young female consumers were found to be willing to purchase used luxury goods when they perceived a high level of green value [48]. Vermeir and Verbeke [49] found that consumer effectiveness perceptions were positively related to consumers’ intention to buy organic food. It was found that young female consumers have an intention to purchase luxury goods when recognizing a high level of green value [50].

**Hypothesis** **3** **(H3).***Perceived consumer effectiveness will have a positive effect on the purchase intention of eco-friendly products*.

#### 2.4.2. Intention to Purchase and Eco Product Purchase Behavior

It has long been understood that purchase intention is crucial to understanding, interpreting, predicting, and influencing consumer behavior. However, in the context of ethical consumerism, it is not well-understood that purchase intention predicts purchase behavior [35]. Therefore, to investigate consumers’ purchasing behavior of eco-friendly products, it is necessary to examine the relationship between purchase intention and purchase behavior. Wee et al. [51] investigated the correlation between consumers’ perception, purchase intention, and actual purchase behavior of organic foods based on the planned behavioral theory, and it was identified that the actual purchase behavior of organic foods was significantly influenced by the purchase intention of products.

**Hypothesis** **4** **(H4).***The purchase intention of eco-friendly products will have a positive effect on the purchase behavior of eco-friendly products*.

#### 2.4.3. Moderating Effect of Situational Context

Richter and Klöckner [52] studied the relationship between consumer knowledge, attitude, intention, and consumption behavior in consuming eco-friendly seafood. It was assumed that habits, situational conditions (ease of purchase, price premium, label, availability, etc.), and socioeconomic conditions (age, income, education, etc.) had a moderating effect on the relationship between intention and behavior. Analysis results revealed that habits weakened the relationship between intention and responsible consumption behavior. Even if consumers continue to have a positive attitude toward seafood consumption, a lack of trust in certification bodies (e.g., the use of seafood labels) is an obstacle to forming specific intentions. Grimmer and Miles [45] identified a gap between consumers’ intention to purchase environmentally friendly products and actual purchasing behavior, and suggested that contextual factors weakened the relationship between intention and behavior.

Vermeir and Verbeke [49] confirmed that a high ease of purchase has a positive relationship with attitude and intention for purchasing sustainable products. The study also identified that ease of purchase can act as a barrier to making sustainable consumption decisions. Consumers who thought that eco-friendly products were in short supply said that they could not purchase products despite their positive attitude toward the product. Grimmer et al. [18] stated that the ease of purchase can play a moderating role in the relationship between intention and behavior. The results revealed that ease of purchase had a moderating effect on the relationship between intention and actual behavior.

**Hypothesis** **5** **(H5).***Ease of purchase moderates the relationship between purchase intention and purchase behavior of eco-friendly products*.

Joshi and Rahman [7] stated that producers and marketers should not only launch products with eco labels, but also strive to build consumer confidence in eco labels, and the government should monitor the reliability of messages published on them. Moussa and Touzani [53] presented quality labels as a signal to reduce problems with asymmetric information when consumers were willing to purchase products. The perceived reliability of the label significantly affects the quality perception of the product, and consequently, impacts the purchase intention of the product. Accordingly, it was argued that the reliability of the label was significant.

**Hypothesis** **6** **(H6).***Eco label credibility moderates the relationship between purchase intention and purchase behavior of eco-friendly products*.

## 3. Methodology

### 3.1. Measure Development

The empirical data for this study were collected through a paper-based survey in South Korea. We distributed questionnaires to visitors of eco-friendly stores at super supermarket (SSM), and immediately collected them after respondents responded. For the sample of this study, we set the population of this study as adult men and women with experience in purchasing eco-friendly products. Moreover, we obtained 236 responses through convenience sampling. After eliminating insincere and incomplete responses through data filtering, we finally obtained a total number of 220 usable responses. Table 1 presents the respondents’ demographic details. The questionnaire, which first defined eco-friendly products for the respondents’ benefit, included questions that measured environmental consciousness, purchase intention, and purchase behavior of eco products, ease of purchase, label credibility, and demographic characteristics of the consumers. With the exception of demographic questions, items were measured using a 7-point Likert scale (1: not at all~7: totally agree), which indicated the degree of agreement with each of the items. All measurement items were modified and supplemented according to this study based on questions that secured the reliability and validity from previous studies.

As a measure of environmental consciousness, three questions used by Paul, Modi, and Patel [23] were used for evaluating environmental concern; three questions used by Joshi and Rahman [26] were used for environmental knowledge; and three questions used by He and Zhan [28] were used for consumer effectiveness. For the measure of purchase intention of eco-friendly products, three questions used by Paul et al. [23] and two questions used by Cleveland et al. [54] were used for the purchase behavior of eco-friendly products. Three questions previously used by Gleim et al. [30] were used to measure ease of purchase, and five others from previous studies [53,55,56] to measure label credibility. After developing a pre-measurement scale based on previous research, a pre-test was first conducted with 40 undergraduate and graduate students who had purchased eco-friendly products in order to derive measurement items. We created metrics based on our preliminary test results. Appendix A lists the construct measuring items. 

### 3.2. Measurement Model

A structure equation model approach was used in this study. First, we conducted a confirmatory factor analysis to test the validity of the constructs. Then, the Cronbach’s alpha was calculated for each latent variable, which consists the remaining observed variables. Gefen et al. [57] recommend the use of the internal consistency coefficient or internal consistency reliability. Cronbach’s alpha coefficients were all higher than the reference value of 0.6, satisfying the appropriate level of internal consistency. The item-to-concept average correlation coefficient also exceeded the reference value of 0.6. Additionally, the value of the factor loading value was more than 0.5, which is the reference value; the value of the reliability coefficient was more than 0.6, which is the reference value; and the variance extraction value was more than 0.5, which confirmed the convergence validity of the constituent concept [58]. 

Table 2 confirms the results of factor analyses on the reliability and validity of these specific measurement items. As a result of verifying the overall suitability of the entire model, X^2^ was 508.35 (df = 338) and the p value was 0.000. GFI was 0.91 above the recommended level (>0.90), and AGFI was 0.88 above the recommended level (>0.80). RMSEA was 0.032, which was below the recommended level of 0.08. NFI was 0.94, CFI was 0.98, and IFI was 0.98, which was found to meet the recommended level (>0.90) [59].

Table 3 shows the results of analyzing the discriminant validity. We calculated the square root of each factor’s AVE and its correlation coefficients with other factors. As a result of the analysis, it was confirmed that there is validity for discrimination between notions because the square root of each factor’s AVE is larger than its corresponding correlation coefficients with other factors as shown in the presented table. All fit indices are acceptable [60]. Thus, the results indicate an adequate model fit between our research model and the empirical data.

## 4. Results

For the hypothetic SEM model, we used SPSS 22.0 and AMOS 22.0 to test whether the empirical data conformed to the proposed model. The model included twenty-two items describing seven latent constructs. We examined the model fit of our research, as shown in Table 4. The common criteria in the SEM were suggested by Hair et al. [61]. All fit indices of this study are acceptable. 

Table 5 shows the results of hypothesis verification on the relationship between the level of environmental consciousness and the purchase intention of eco-friendly products. H1 to H3 predicted that environmental consciousness would affect one’s purchase intention of an eco product. The results showed that environmental knowledge significantly influenced purchase intention (β = 0.42, *p* < 0.001), and perceived consumer effectiveness also significantly affected purchase intention (β = 0.28, *p* < 0.01), but environmental concern did not significantly affect purchase intention (β = 0.18, n.s). This finding supports H2 and H3. H4 posited that one’s purchase intention of an eco product affects purchase behavior. The results show that purchase behavior toward eco products was significantly influenced by purchase intention (β = 0.91, *p* < 0.001), indicating the support of H4. We further analyzed the effect size. Effect size indicates whether a structure has a real impact on other structures. The generally recommended values are 0.02, 0.15, and 0.35, respectively [62]. The effect size of the relationship between environmental knowledge and purchase intention was 0.044, and between perceived consumer effectiveness and purchase intention was 0.067.

Table 6 and Table 7 show the results of hypothesis verification on the moderating effects of situational factors in the relationship between purchase intention of eco-friendly products and the actual purchase behavior of eco-friendly products. To verify the moderating effect of situational factors, an analysis of the difference between the two groups using a structural equation model was conducted. Upon examining the difference in the kai square between groups in order to investigate the moderating effect of label trust, H5 was △x^2^ = 6.59, which indicates that the statistically marked difference at the significance level of 0.05 and the high ELC are notably higher than the low ELC. This result shows that eco label credibility has a positive moderating effect on the relationship between the purchase intention and the behavior toward eco products. Hypothesis 5 is therefore supported. That is, the higher the eco label credibility, the higher the influence of purchase intention on the behavior toward eco products. Upon examining the difference in kai square between groups to investigate the moderating effect of ease of purchase, H6 was Δx^2^ = 6.89, indicating a statistically significant difference at the significance level of 0.05. This also means that ease of purchase has a positive moderating effect on the relationship between purchase intention and behavior toward eco products. Hypothesis 6 is thus supported. In other words, the higher the ease of purchase, the stronger the effect of purchase intention on behavior.

## 5. Discussion

This study indicated that existing studies on eco-friendly products had limitations in predicting purchase behavior by measuring only purchase intention, and sought to examine the relationship between purchase intention and purchase behavior by expanding on actual purchase behavior. It was intended to examine the moderating effect depending on situational factors, believing that there would either be a barrier or promotion of situational factors for the phenomenon whereby eco-friendly products are not actually purchased despite real purchase intentions. Additionally, owing to the recent increase in environmental problems, consumers’ environmental consciousness will impact the purchase of eco-friendly products, and the study attempted to examine the relationship between environmental consciousness and eco-friendly product purchase.

The analysis results find that environmental knowledge and perceived consumer effectiveness factors act as independent antecedents of the purchase intention of eco product. *Although some researchers have investigated the factors affecting the antecedents of purchase intentions for green products via the mediating role of attitude [13,63,64], this study found that two factors act as the antecedents of eco-friendly purchase intention, which are directly based on an intention–behavior model of eco-consumer behavior*. Among environmental consciousness factors, environmental knowledge has a stronger effect on the purchase intention of eco products than PCE. *This finding implies that consumers who are interested in eco-friendly products and are more knowledgeable on the matter are more likely to act eco-friendly. Moreover, the more knowledge of environmental problems and issues, the more consumers intend to purchase eco-friendly products. This study also confirmed that PCE is an important predictor of ecologically conscious consumer behavior. Furthermore, eco-friendly consumers are more internally controlled by a belief in the self which contributes to a more action-oriented attitude, rather than a collective effectiveness imposed by society and the government [65]. Therefore, consumers with high PCE have been shown to believe that they could possibly handle ecological issues by themselves with their own efforts*. The results also show that the purchase intention of an eco product is positively associated with the purchase behavior toward an eco product. In addition, this study indicates that ease of purchase and eco label credibility moderates the effect of purchase intention on purchase behavior. This result means that consumers who can easily find and purchase eco-friendly products around them are relatively more likely to buy eco products. *Furthermore, these results also provide contributions to previous studies on the gap between intention and behavior in green consumerism. Previous studies related to green consumption have used the role of cognitive view to explain the gap between intention and behavior [16,35,43]. Under this view, studies usually consider the intention as highly correlated with behavior. However, our study focuses on the situational context to examine the moderators that help close the gap between intention and behavior in green consumption. Therefore, the findings of our study have shown that there always exists at least a gap between the intention and the behavior in eco-friendly consumption.* In addition, this finding also supports previous studies [18,26] which conclude that the more favorable situation increases the translation of intentions to behavior. Another finding of this result means that a higher eco label credibility strengthens consumers’ behavior toward eco products.

### 5.1. Theoretical Implications

The following theoretical implications are presented based on the results of this study. First, from an environmental perspective, the variables affecting the purchase intention of eco-friendly products were considered as consumers’ environmental consciousness divided into environmental interest, environmental knowledge, and consumer effectiveness perception. The effect on the relationship of eco-friendly product purchase intentions was confirmed. The dimensions of environmental consciousness, environmental knowledge, and consumer effectiveness perception were identified as variables affecting the purchase intention of eco-friendly food—this differs from the research results of Maichum et al. [47] in that it does not directly affect the purchase intention of eco-friendly products. However, consistent with Frick et al. [27], environmental knowledge and the perception of effectiveness directly affect behavioral intentions. This study identified that environmental interest did not affect the purchase intention of eco-friendly food. These results show that environmental interest cannot be considered a factor that directly affects the purchase intention of eco-friendly food, and other factors are required as parameters. Newton et al. [4] stated that more information was required to support purchase decisions before converting to purchase intention of eco-friendly products. Therefore, it would be more meaningful to examine an interest in health as a parameter in the relationship between an interest in the environment and the intention to purchase eco-friendly products.

Second, it was judged as insufficient to predict purchase behavior by measuring only consumers’ purchase intention, owing to the characteristics of eco-friendly food, and the purchase behavior of eco-friendly products was examined as a result variable. Previous studies suggested that in ethical or eco-friendly products, purchase intention did not predict purchase behavior, and purchase intention and purchase behavior were inconsistent. Therefore, it was insufficient to regard purchase intention as a predictor of purchase behavior. Accordingly, in this study, the purchase behavior of eco-friendly products was measured, and it was confirmed that the purchase intention of eco-friendly products was a variable affecting the purchase behavior of eco-friendly products.

Third, owing to the nature of eco-friendly products, there is an intention to purchase products; however, actual purchases have not increased. This phenomenon can be explained as there being another factor impacting the relationship between intention and behavior in purchasing eco-friendly products. However, in previous studies, only theoretical concepts were presented on barriers and catalyst factors for the discrepancy between the intention and behavior of purchasing eco-friendly products. This study presented situational factors to examine the relationship between eco-friendly product purchase intention and eco-friendly product purchase behavior and examined the moderating effects of the suggested situational factors.

### 5.2. Practical Implications

The results of this study provide eco product company managers with insight into how to improve consumers’ purchase behaviors. The predominant implication is that environmental knowledge and consumer effectiveness perception affect the purchase intention of eco-friendly products and lead to eco-friendly product purchase behavior. The findings of this result recommend increasing consumers’ knowledge of eco product performance which can help to accomplish personal goals of environmental impact. This social and educational green consumption strategy would help to not only increase eco-friendly consumption behaviors, but also turn consumer intention into actual green behaviors. Under this strategy, consumers need to be aware of their personal impacts via consuming eco products. Therefore, companies should focus not only on promoting products, but also on delivering knowledge about the environment so that consumers can augment their environmental knowledge. It can also be predicted that the sales of eco-friendly products will increase in the future as information on how consumer behavior will affect the environment is steadily shared with the public. Additionally, Korean consumers are still more interested in their own health than in the ecological environment when deciding on the purchase of eco-friendly products. Therefore, when advertising eco-friendly products, it will be more effective to emphasize both an interest in the environment and health for the consumers themselves, as well as their families. In the future, it is expected that a promotion concept for consumption promotion linking health and environmental protection will be required.

Another implication of our finding is related to the situational context. It is necessary to focus on the credibility of the eco label and the ease of purchase to facilitate sales of eco-friendly products. However, it would appear that the mere presence of eco labels does not necessarily drive product credibility. The brand managers of green products should also consider which eco label to use, as there are many in the marketplace. As eco labels interact with brand or product evaluation, selecting the right eco label is even more important for brand managers.

Additionally, eco-friendly food companies should make it easier for consumers to purchase products. Affordability can impact the purchase of eco-friendly foods, such as organic foods, and this is not under the consumer’s control. Supply chains determine the availability of eco-friendly food to consumers. According to a 2017 survey by the Ministry of Agriculture, Food and Rural Affairs, most consumers (59.3%) have difficulty buying eco-friendly foods at large discount stores, SSM (corporate supermarkets), and local supermarkets, except for specialty stores. Most consumers buy eco-friendly foods at large discount stores, but there are occasions when they do not have eco-friendly foods in stock, and products that they usually buy are often sold out. Therefore, eco-friendly food management in large marts should be thoroughly managed through smooth communication between eco-friendly food producers and sellers so that various items can be supplied steadily, and finally, inventory management should be improved. It is also necessary to introduce measures to facilitate the purchase of these eco-friendly products.

## 6. Limitations and Future Studies

Although this study reveals several theoretical and practical implications, it has some limitations. This study examined a sample of a relatively high percentage of environmentally aware consumers. Future research could include the differences between green and non-green consumer responses. 

Consumers’ income and cultural backgrounds might influence their decision making process. However, the present study was based on a sample in only one developed country. Therefore, it should be replicated in other countries to understand how different consumers associate their perceptions and outcomes. Moreover, intention and behavior are measured at the same point in time throughout the same sample. Future research may apply the survey at different moments of time so as to better determine the intention–behavior gap. As such, various characteristics of the sample and cross-sectional research will contribute to the generalization of the research result. 

This study attempted to present new influencing variables based on the planned behavioral theory of Fishbein and Ajzen, but it could not examine the variables that affect the existing purchase intention. It would be more meaningful to examine various subjective norms, perceived control, and attitude variables proposed by Fishbein and Ajzen.

Another possible future extension of this work could be to investigate whether the results from this study can be applied to various eco label types including color, size, and sponsor.

Similarly to previous researches [66,67], this study has the possibility of a social desire ability bias by using a Likert scale. Some respondents normally lie in the questionnaire due to biasness, hence leading to wrong conclusions. Bias between self-reported behavior and real behavior might be overcome in future research via an experimental study comparing real behaviors with previous intentions.

As an approach to resolve the discrepancy between purchase intention and purchase behavior toward eco-friendly products, this study focused on situational factors that can control the effect of purchase intention on behavior. In future studies, it will be necessary to examine the moderating effects of demographic, socioeconomic, and socio-psychological variables.

## Figures and Tables

**Figure 1 ijerph-20-05312-f001:**
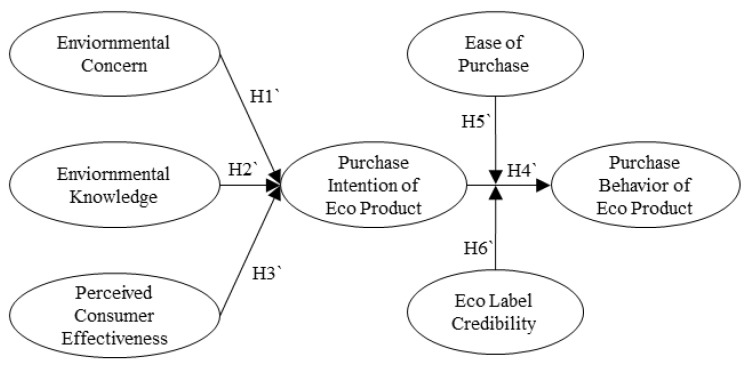
Research model.

**Table 1 ijerph-20-05312-t001:** Profile of respondent characteristics.

Demographics	Item	Subjects	
		Frequency	Percentage
Gender	Male	114	51.8
	Female	106	48.2
Age	20–29	40	18.2
	30–39	119	54.1
	40–49	32	14.5
	>49	29	13.2
Marriage	Married	104	47.3
	Single	116	52.7
Monthly income	<3,000,000 Won	65	29.5
	3,000,000–<4,000,000 Won	35	15.9
	4,000,000–<5,000,000 Won	28	12.7
	5,000,000–<6,000,000 Won	24	10.9
	6,000,000–<7,000,000 Won	20	9.1
	7,000,000–<8,000,000 Won	17	7.7
	≥8,000,000 Won	31	14.1
Education level	High school or below	31	14.1
	College	167	75.9
	Graduate school or above	22	10.0

**Table 2 ijerph-20-05312-t002:** Item loadings and reliabilities.

Construct	Item	Factor Loading	CR	AVE	Cronbach’s Alpha
Environmental concern	ENC1	0.727	0.874	0.701	0.822
	ENC2	0.891			
	ENC3	0.736			
Environmental knowledge	ENK1	0.801	0.753	0.507	0.858
	ENK2	0.926			
	ENK3	0.741			
PCE	PCE1	0.844	0.823	0.608	0.859
	PCE2	0.821			
	PCE3	0.798			
Eco label credibility	ELC1	0.874	0.904	0.652	0.953
	ELC2	0.890			
	ELC3	0.863			
	ELC4	0.837			
	ELC5	0.821			
Ease of purchase	EOP1	0.522	0.779	0.540	0.778
	EOP2	0.512			
	EOP3	0.778			
Purchase intention	INT1	0.809	0.897	0.743	0.937
	INT2	0.830			
	INT3	0.814			
Purchase behavior	BEH1	0.833	0.744	0.678	0.862
	BEH2	0.897			

**Table 3 ijerph-20-05312-t003:** Correlation coefficient matrix and roots of the AVEs.

	1	2	3	4	5	6	7
1. Environmental concern	**0.84**						
2. Environmental knowledge	0.15	**0.71**					
3. PCE	0.49	0.42	**0.78**				
4. Eco label credibility	0.09	0.27	0.43	**0.81**			
5. Ease of purchase	0.13	0.11	0.08	0.04	**0.73**		
6. Purchase intention	0.27	0.51	0.47	0.50	0.39	**0.86**	
7. Purchase behavior	0.26	0.47	0.27	0.27	0.24	0.14	**0.82**

Numbers in the diagonal in the bold values present the square root of AVE.

**Table 4 ijerph-20-05312-t004:** Summary of fit indices.

Fit Indices	$\chi^{2}/\mathrm{df}$	GFI	AGFI	NFI	CFI	RMSEA
Recommended value	<3	>0.90	>0.80	>0.90	>0.90	<0.08
Value in this study	1.51	0.96	0.94	0.96	0.98	0.04

**Table 5 ijerph-20-05312-t005:** Results of hypotheses test.

Hypothesis	Path	Coeff.	t-Value	Result
H1	ENC-INT	0.18	1.62	Not Supported
H2	ENK-INT	0.42	7.03	Supported
H3	PCE-INT	0.28	3.36	Supported
H4	INT-BEH	0.59	9.10	Supported

**Table 6 ijerph-20-05312-t006:** Result of moderating effect of Hypothesis 5.

Hypothesis	High ELC		Low ELC		Δx^2^	Result
	Path	t	Path	t		
H5	0.81	7.02	0.44	5.09	6.59 *	Supported

* *p* < 0.05, ELC: Eco label credibility.

**Table 7 ijerph-20-05312-t007:** Result of moderating effect of Hypothesis 6.

Hypothesis	High EOP		Low EOP		Δx^2^	Result
	Path	t	Path	t		
H6	0.77	8.54	0.42	4.46	6.89 *	Supported

* *p* < 0.05, EOP: Ease of purchase.

## Data Availability

Not applicable.

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
