# Peer review of "Environmental Consciousness, Purchase Intention, and Actual Purchase Behavior of Eco-Friendly Products: The Moderating Impact of Situational Context"

_ijerph, 2023, doi:10.3390/ijerph20075312_

Round 1

Reviewer 1 Report

1.       Line 25- ‘owing to increase in fine dust’ sounds strange, did you mean air pollution?

2.       First paragraph of introduction needs proofreading for the grammar to flow better.

3.       Lines 40-42 and 68-72 are both worded like the aim/objective of the paper. It would sound better to present them together. Perhaps as ‘This study aims to 1) …. and 2) ….’.

4.       Line 45- reference for the P&G study needed

5.       Line 59- Spelling of author Josh does not match reference section.

6.       Line 88- Suggest ‘previous research’ instead of ‘previous studies’ to flow better with ‘this study identified…’

7.       Line 93- Is this also the aim of the study? It should be at the end of the introduction with other statements (comment 3).

8.       Section 2 is well-written, but highlights the weaknesses of the writing of the introduction. I suggest both authors work to match the writing styles of the two sections.

9.       Line 260- From which country/ area were the respondents recruited from? What was the distribution method?

10.   Line 266-267- Do you mean ‘Totally agree’?

11.   Lines 270-27- Can you share the questions in a table or in the supplementary data?

12.   Line 308-311- Can there be a short explanation for hypotheses 1, 2, and 3 (lines 303-305) like hypothesis 4?

13.   Line 324- Your study didn’t indicate that other had limitations. That was determined with the literature review. I suggest rewording that part of the sentence.

14.   Remove line 377. Remove ‘Firstly’ and ‘Secondly’ in that section.

15.   Line 412- I suggest ‘This study has some limitations.’ Consider rewording to exclude First, Second, Third at the beginning of paragraphs.

16.   No conclusion section?

Author Response

Thank you for your time and comments that helped improve our article. We have carefully considered your comments and recommendations and used them to improve the quality of the paper. The responses to your advices are below.

Below we address each of your comments in italics in turn and explain our responses. Other changes were also reflected in the manuscript. Paragraphs or sentences with major changes made to the original paper were shown in colors other than black.

Once again, authors appreciate you for your careful and fruitful comments directions for revision. Please be kind to read below.

Reviewer 2 Report

Abstract

I suggest adding and describing in brief the research methodology, population and data analysis technique used in this study.

Introduction

- Kindly define the abbreviation once being introduced for the first time.

- Need to elaborate the purpose of the study in one paragraph.

- At the end of this section, add paragraph shows the structure of the paper. 

Theoretical Foundation: I didnt see this important section. Kindly write theoretical background and support with any relevant theory. The authors should use a base-line theory to support the relationships between the variables.

Data Collection Method/Procedure: 

In the Methodology part, the methodology suffers from some technique issues. I think if the authors want results to be convincing, the methodology should be further developed.

What is the Techniques of Data Analysis, it need more explanation.

Tables 5 and 6 show the results of hypothesis verification on the moderating effects are missing, kindly add them.

Discussion is too shallow. Need more discuss and could connected with previous studies.

Limitations:  Kindly correct this section heading to be Limitations and Future Studies. In addition, I would recommend the authors add some more recommendations; the presentation of future work could be detailed.

References: The number of references cited are few. I would recommend to add more related research particularly the literature review and discussion sections.

I suggest additional reading that will surely help increase the study’s general impact, taking into consideration the international perspective. References (to be consulted and not necessarily to be cited):

1- How Do Perceived Value and Risk Affect Purchase Intention toward Second-Hand Luxury Goods? An Empirical Study of U.S. Consumers.

2- A sensemaking perspective on the association between social media engagement and pro-environment behavioural intention.

3- Assessing customers perception of online shopping risks: A structural equation modeling–based multigroup analysis.

Good luck

Author Response

(The authors gave the same response as above.)

Reviewer 3 Report

Thank you for the opportunity to read the article titled Environmental Consciousness, Purchase Intention, and Actual Purchase Behavior of Eco Friendly Products: The Moderating Impact of Situational Context  in which authors attempt to explore consumers' environmental consciousness as a factor influencing the purchase of eco-friendly products, and investigated situational factors that induce hesitancy in purchasing eco-friendly products. While the paper is mostly well-written and structured, however, several shortcomings remain which the authors need to address before this manuscript can be accepted for publication. My comments and suggestions for the authors are as follows:

Abstract

 The abstract could benefit from a clearer statement of the research question or problem being addressed. The methodology used in the study should be briefly described. The results should be stated more clearly and their implications more explicitly drawn. The language used should be more precise and professional. Additionally, the abstract should be written in complete sentences.

Introduction

¾    The introduction could be more concise and focused.

¾    The background information on environmental awareness and eco-friendly products could be presented in a clearer and more structured manner.

¾    The literature review on the gap between purchase intention and purchase behavior could be more comprehensive and recent.

¾    The purpose of the study could be stated more explicitly and clearly.

¾    The contribution of the study to the existing literature could be highlighted.

Literature Review and Hypothesis Development

The section on Literature Review and Hypothesis Development is well organized and presents the concept of environmental consciousness and its components. However, it could benefit from further improvement in the following ways:

¾    Definition of environmental consciousness could be more clear and concise.

¾    The authors could provide more examples or case studies to better illustrate the different dimensions of environmental consciousness and how they impact eco-friendly purchasing behavior.

¾    The relationship between environmental consciousness and eco-friendly purchasing behavior could be more clearly defined and explained, including any potential limitations and challenges.

¾    The section on Situational Context could be enhanced by including a clearer explanation of the various factors that impact eco-friendly purchasing behavior, and how they interact with environmental consciousness.

¾    The authors could also discuss the current state of research in this area and identify any gaps in the literature that this study aims to fill. I suggest the authors to include the following recent literature:

https://doi.org/10.3390/ijerph18041912

https://doi.org/10.3389/fenvs.2022.917984

https://doi.org/10.1016/j.landusepol.2020.105250

Methodology

The methodology section of the study provides a comprehensive description of the measurement process of the study's variables. However, it can be improved in the following ways:

¾    More detailed explanation of the sources of the questions used for each measure, including the specific studies, could help in establishing the robustness of the measures.

¾    A clearer explanation of the modification and supplementing of the questions used for each measure to secure reliability and validity could enhance the transparency of the study's methods.

¾    More detail on the factor analysis process, including how the internal consistency of each measure was established, could improve the rigor of the methodology section.

¾    A brief explanation of the reference values used in the study, such as 0.6 for the internal consistency and 0.5 for the factor loading, could add clarity to the process used to establish reliability and validity.

¾    A discussion on the limitations of using the Likert scale and its impact on the results of the study could help to contextualize the findings and provide a more nuanced interpretation.

¾    In the section on measurement model, more detail on the significance of the values of X², GFI, AGFI, RMR, NFI, NNFI, CFI, and IFI and how they were used to evaluate the overall suitability of the entire model could enhance the depth of the methodology section.

Results

The results section of the study presents the findings of hypothesis testing on the relationship between environmental consciousness, purchase intention, and actual purchase behavior of eco-friendly products. There are a several limitations and areas for improvement.

¾     The first shortcoming is that there is a lack of explanation of the sample size and demographic characteristics of the participants. This information is important to understand the generalizability of the results to the population.

¾     Another limitation is that the results only show the statistical significance of the relationships between the variables, but not the magnitude of the effects. It would be helpful to provide information on the effect sizes, such as Cohen's d or omega squared, to better understand the practical significance of the results.

¾     There is also a lack of discussion on potential confounding variables or control for potential confounding factors that may have affected the results. This could include socio-demographic factors, personality traits, or other consumer behavior-related factors.

¾     Finally, this section is very short. Authors should add more results to improve the quality of the manuscript and make it more comprehensive.

Discussion

¾     The discussion section does a good job of summarizing the key findings of the study and the theoretical and practical implications of those findings.

¾     Authors should add a clearer presentation of the relationship between environmental knowledge, consumer effectiveness, and eco-friendly product purchase intention, as well as the relationship between situational factors and eco-friendly product purchase behavior.

¾     This section could benefit from a more thorough explanation of the limitations of previous studies on eco-friendly products, and how the current study addresses those limitations.

¾     Authors should consider possible confounding variables and alternative explanations for the results, and address the generalizability of the findings.

¾     Finally, please add a clearer articulation of the implications for companies, including how they can use the findings to promote eco-friendly products more effectively.

Author Response

(The authors gave the same response as above.)

Round 2

Reviewer 2 Report

Kindly see attached file

Author Response

Once again, authors appreciate our reviewer for their careful and fruitful comments directions for revision. We appreciate if the reviewer read once again the revised manuscript, which has been significantly revised. 

Reviewer 3 Report

I have no further comments.

Author Response

Authors thank you for your constructive and thoughtful comments. These have helped to improve the exposition of the paper and enhance its overall quality.

All the best.